# Molecular-Based Treatment Strategies for Osteoporosis: A Literature Review

**DOI:** 10.3390/ijms20102557

**Published:** 2019-05-24

**Authors:** Yuichiro Ukon, Takahiro Makino, Joe Kodama, Hiroyuki Tsukazaki, Daisuke Tateiwa, Hideki Yoshikawa, Takashi Kaito

**Affiliations:** Department of Orthopedic Surgery, Osaka University Graduate School of Medicine, 2-2 Yamadaoka, Suita, Osaka 565-0871, Japan; yuichiro-ukon721@umin.ac.jp (Y.U.); t-makino@za2.so-net.ne.jp (T.M.); joekodama@gmail.com (J.K.); tsukazaki.hiroyuki@gmail.com (H.T.); tateiwa.daisuke1@gmail.com (D.T.); yhideki@ort.med.osaka-u.ac.jp (H.Y.)

**Keywords:** osteoporosis, bone metabolism, osteoblast, osteoclast, bisphosphonate, parathyroid hormone, calcitonin, receptor activator of nuclear factor κB, sclerostin, stem cell

## Abstract

Osteoporosis is an unavoidable public health problem in an aging or aged society. Anti-resorptive agents (calcitonin, estrogen, and selective estrogen-receptor modulators, bisphosphonates, anti-receptor activator of nuclear factor κB ligand antibody along with calcium and vitamin D supplementations) and anabolic agents (parathyroid hormone and related peptide analogs, sclerostin inhibitors) have major roles in current treatment regimens and are used alone or in combination based on the pathological condition. Recent advancements in the molecular understanding of bone metabolism and in bioengineering will open the door to future treatment paradigms for osteoporosis, including antibody agents, stem cells, and gene therapies. This review provides an overview of the molecular mechanisms, clinical evidence, and potential adverse effects of drugs that are currently used or under development for the treatment of osteoporosis to aid clinicians in deciding how to select the best treatment option.

## 1. Introduction

At the Consensus Development Conference in 1993, osteoporosis was defined as a systemic skeletal disease characterized by low bone mass and microarchitectural deterioration of bone tissue, with a consequent increase in bone fragility and susceptibility to fracture [1]. The World Health Organization Study Group has proposed general criteria for the diagnosis of osteoporosis based on dual-energy X-ray absorptiometry, which is considered the standard tool for the evaluation of osteoporosis. According to the criteria, osteoporosis is diagnosed when the bone mineral density (BMD) is 2.5 or more standard deviations below the young-adult mean [2,3]. The incidence of osteoporosis increases with aging. In addition, the population aged 60 years or older has been continuously growing in many countries [4]. More than 75 million people have osteoporosis in Europe, Japan, and the USA alone [2]. Worldwide, it is estimated that 9 million people suffer from osteoporotic fractures each year [5]. Therefore, treatments for osteoporosis with various medicines have been receiving increasing attention globally.

Osteoporosis is caused by an imbalance in bone remodeling, which is an ongoing process in which mature bone tissue is removed by osteoclasts (bone resorption) and new bone tissue is formed by osteoblasts (bone formation). Excessive bone resorption or inadequate new bone formation during bone remodeling can result in osteoporosis [6]. To maintain bone homeostasis, osteoblast and osteoclast functions are coordinated by a wide variety of molecules (Figure 1).

Osteoblasts originate from mesenchymal stem cells (MSCs). Runt-related transcription factor 2 (Runx2), also known as core-binding factor subunit alpha-1, is a key transcription factor for osteoblast differentiation from MSCs and pre-osteoblasts. Runx2 expression is the first requisite step in the determination of osteoblast commitment, followed by expression of Sp7 and Tcf7, which are also essential for osteoblastic differentiation [7]. Runx2 is regulated by multiple signals, such as bone morphogenetic proteins (BMPs) and Wnt/β-catenin. As for BMP signaling, SMAD1/5/8 are phosphorylated particularly by BMP2 and BMP4, and finally, stimulate osteoblast differentiation by activating Runx2. Regarding Wnt/β-catenin signaling, Wnt proteins (particularly Wnt3a and Wnt10b) bind to Frizzled and lipoprotein receptor-related protein (LRP)-5/6 receptors, and consequently increase Runx2 levels through either β-catenin stabilization or protein kinase Cδ [8,9]. Sclerostin and dickkopf (DKK)-1 inhibit the Wnt/β-catenin pathway through the LRP-5/6 receptor, thus leading to a decrease in Runx2 expression.

Osteoclasts differentiate from hematopoietic stem cells through monocyte/macrophage lineage upon stimulation with monocyte/macrophage colony-stimulating factor and activation of receptor activator of nuclear factor κB (RANK) by its ligand (RANKL) [10]. RANKL secreted by osteoblasts and osteocytes binds to RANK on osteoclast precursor cells, which eventually differentiate into osteoclasts. Osteoprotegerin (OPG), which is also produced by cells in the osteoblast lineage, is a soluble decoy receptor of RANKL that prevents binding of RANKL to RANK [11]. The RANKL-RANK-OPG interaction plays an essential role in bone homeostasis through osteoclast regulation [12].

Osteocytes, which are completely embedded in the bone matrix, descend from MSCs through osteoblast differentiation and orchestrate bone remodeling by regulating osteoblasts and osteoclasts. Especially, they exclusively secrete sclerostin [13].

Treatments for osteoporosis are divided into several groups, including therapies based on essential nutrients, anti-resorption drugs, anabolic drugs, and combinations of these. Major essential nutrients for the treatment of osteoporosis are calcium, vitamin D, and vitamin K2, all of which are involved in bone metabolism. Anti-resorptive drugs, which suppress bone resorption, include calcitonin, estrogen and selective estrogen receptor modulators (SERMs), bisphosphonates (BPs), and anti-RANKL antibody. Anabolic drugs, which enhance bone formation, include parathyroid hormone (PTH) and sclerostin inhibitors. Additionally, stem cell therapies for osteoporosis have been receiving increased attention in recent years.

These therapies are helpful for the treatment of osteoporosis as evidenced by numerous clinical trials. However, nearly all these therapies have side effects because of the long-term drug administration for osteoporosis treatment. In this literature review, we summarize the mechanisms of action of current and anticipated drugs in terms of basic bone biology, major clinical trials, and side effects, to aid clinicians in deciding how to select the best treatment option (Table 1).

## 2. Calcium

### 2.1. Mechanism of Action

Calcium is a fundamental bone mineral but tends to be in short supply in the diet. Low serum calcium level will raise PTH secretion, which eventually causes a high bone turnover. Conversely, administration of calcium will reduce PTH release and eventually suppress bone resorption. However, calcium is a threshold mineral, that is, surplus ions are excreted. Excessive intake of calcium, therefore, has no benefit for bone health. Therefore, calcium administration is best used only in patients whose pathology of osteoporosis is directly associated with calcium shortage, or patients with secondary hyperparathyroidism. Hence, in most cases, clinical application of calcium for osteoporosis is supplementary to bisphosphonates or anti-RANKL drugs.

### 2.2. Clinical Trials for the Treatment of Osteoporosis

Calcium seems to have somewhat positive effects on BMD, but the efficacy of calcium in preventing osteoporosis-related fractures remains controversial. According to a meta-analysis by Shea et al. [14], treatment with calcium can only slightly increase the BMD as compared with placebo. The pooled difference in percentage change from baseline was 2.05%. Reit et al. [15] reported similar findings by the trial in 1471 healthy postmenopausal women. Calcium supplementation had a positive effect on total BMD, with the difference between calcium and placebo groups at 5 years of 1.2%.

Tang et al. [16] reported, by meta-analysis, that calcium treatment was associated with a 12% risk reduction in fractures (relative risk (RR), 0.88; 95% confidence interval (CI), 0.83–0.95). Several reports showed that calcium reduces the incidence of vertebral fractures, whereas it does not prevent non-vertebral fractures [17,18,19,20,21]. In contrast, some studies showed that calcium has no effect on any osteoporotic fractures [22,23]. Long-term compliance may be a factor that limits the efficacy. Prince et al. [24] reported that calcium has a positive effect on fracture prevention if the compliance rate is higher than 80%.

Recent studies have focused on the combination of calcium and vitamin D. A meta-analysis showed a significant 15% reduction in summary relative risk estimates of total osteoporotic fractures [25]. In contrast, Kahwati et al. [26] concluded by systematic review that this combination was not associated with reduced total fracture incidence (one randomized controlled trial (RCT) (*n* = 36,282); absolute risk difference (ARD), −0.35 %; 95% CI, −1.02% to 0.31%) and hip fracture incidence (two RCTs (*n* = 36,727); ARD from the larger trial, −0.14%; 95% CI, −0.34% to 0.07%).

### 2.3. Adverse Events

The most frequent side effects of calcium are gastrointestinal disorders. Constipation is the major symptom, in which case careful dose adjustment is needed. Hypercalcemia is usually caused by the combination of calcium and vitamin D, and thus, monitoring of the serum calcium level is even more important when the two drugs are taken together. Furthermore, a recent systematic review showed a significant increase in the incidence of urinary stones in case of the combined use of calcium and vitamin D (3 RCTs (*n* = 39,213); pooled ARD, 0.33%; 95% CI, 0.06% to 0.60%), but not when calcium was used alone (three RCTs (*n* = 1259); pooled ARD, 0.00%; 95% CI, −0.87% to 0.87%) [26].

The relation between calcium administration and cardiovascular events, such as cerebral and myocardial infarction, has not been clarified to date. Bolland et al. [66] reported increased risks for cardiovascular events based on a meta-analysis (RR, 1.16; 95% CI, 1.02–1.32). Conversely, Lewis et al. [67] found no difference in the risk of cardiovascular events between calcium supplementation and placebo groups by a 5-year RCT (multivariate-adjusted hazard ratio, 0.938; 95% CI, 0.690–1.275). However, to our knowledge, no RCT specifically designed to investigate this issue has been conducted. Bolland et al. [68] suggested in their recent review that, while calcium supplements have a low risk of major and minor side effects, they have limited benefits in the prevention of osteoporotic fractures.

## 3. Vitamin D

### 3.1. Mechanism of Action

Vitamin D_3_ is the most important among vitamin D forms, which are a group of lipid-soluble secosteroids in the human body. The final metabolite of vitamin D_3_, calcitriol (1α,25-dihydroxyvitamin D_3_), binds to the intranuclear vitamin D receptor in the intestines, bones, kidneys, and parathyroid gland cells. Vitamin D_3_ modulates calcium metabolism, including intestinal absorption, renal excretion, and bone resorption [69]. Vitamin D can be synthesized in the human skin by a photochemical process. However, the capacity of production decreases with age. The elderly are usually at risk of vitamin D deficiency because of a shortage of dietary intake, reduced mobility, and decreased exposure time to sunshine [70]. Moreover, vitamin D shortage causes atrophy of type II muscle fibers [71], which increases the propensity to fall and the risk of fractures.

### 3.2. Clinical Trials for the Treatment of Osteoporosis

Several reports have elucidated that active vitamin D has positive effects in increasing BMD [27] and preventing vertebral fractures [28,29,30]. Therefore, active forms of vitamin D_3_, including calcitriol, alphacalcidol (1α-hydroxyvitamin D_3_, a prodrug of calcitriol), and eldecalcitol (2β-3-hydroxypropyloxy-calcitriol, an analog of calcitriol that was developed in Japan) are mainly used in clinical trials.

A meta-analysis [29] including 25 trials suggested beneficial effects of vitamin D on the incidence of vertebral fractures (RR, 0.63; 95% CI, 0.45–0.88). An RCT including 489 elderly women [27] suggested positive effects also on BMD at 5 years after treatment. The mean change in total body BMD of the calcitriol group was significantly higher than that of placebo (−1.5% vs. −2.8%). In contrast, Bolland et al. [31,68] concluded that vitamin D supplements have no consistent effects on BMD, and weak and inconsistent effects on reducing the risk of total fractures when used as a monotherapy or in addition to calcium supplements (RR, 0.95; 95% CI, 0.88–1.01), based on a meta-analysis.

Interestingly, a meta-analysis indicated a preventive effect of vitamin D on falling (pooled RR, 0.81; 95% CI, 0.71–0.92), which may result from its beneficial influence on the musculoskeletal system [32]. The same result was reported by systematic review (two RCTs: OR, 0.66; 95% CI, 0.44–0.93) [28]. However, Bolland et al. [68] declined this hypothesis in a recent review of four recent RCTs (RR, 0.98; 95% CI, 0.94–1.02).

### 3.3. Adverse Events

Vitamin D monotherapy appears to be safe as non-skeletal adverse events have not been reported. However, the majority of large RCTs reported an unfavorable risk-benefit profile of calcium with vitamin D. Gastrointestinal side-effects, hypercalcemia, kidney stones, and myocardial infarction seem to weigh out the limited benefits on bone homeostasis [31,68].

## 4. Vitamin K2

### 4.1. Mechanism of Action

Vitamin K is a group of fat-soluble vitamins that includes two types: vitamin K1 (phylloquinone) and vitamin K2 (menaquinone). Menaquinone is mainly synthesized from phylloquinone in the human body; thus, vitamin K1 deficiency generally results in vitamin K2 deficiency [72]. The menaquinone family of K2 homologs is a large series of vitamins containing normally unsaturated isoprenyl side chains that vary in length (menaquinone-n). Menaquinone-4 (menatetrenone) is the most-studied, active form of vitamin K2 and is considered to aid the γ-carboxylation of osteocalcin, which is produced by osteoblasts during bone matrix formation [73]. A high serum level of undercarboxylated osteocalcin, an index of vitamin K insufficiency, reportedly is a risk factor for fractures that is independent of BMD in elderly women, and a risk factor for fractures during treatment with bisphosphonate [74,75]. Administration of menatetrenone would decrease the serum level of undercarboxylated osteocalcin [76].

### 4.2. Clinical Trials for the Treatment of Osteoporosis

Three RCTs showed increased lumbar BMD, with a percentage of change ranging from –0.5% to 1.74%, after administration of vitamin K2 in patients [33,34,35]. According to a meta-analysis of RCTs of postmenopausal women and patients using oral steroids for kidney disease, menatetrenone can prevent vertebral as well as non-vertebral fractures (OR favoring vitamin K2 in vertebral fractures, 0.40 (95% CI, 0.25–0.65); OR in non-vertebral fractures, 0.19 (95% CI, 0.11–0.35)) [36]. Shiraki et al. [34] also reported that menatetrenone can prevent mainly vertebral fractures during two years (10.9% at 12 months in the vitamin K2 group vs 30.3% control group, respectively). A large phase IV RCT in Japan for osteoporotic fractures showed that menatetrenone could decrease the risk of vertebral fracture in patients with at least five vertebral fractures at enrollment by 39% (20.3% in the calcium plus vitamin K2 group vs. 33% calcium-only group, *p* = 0.029) [37]. In a recent review, Iwamoto et al. [38] reported that monotherapy of menatetrenone can modestly increase the lumbar spine BMD, and mainly reduces the incidence of vertebral fractures.

### 4.3. Adverse Events

Vitamin K2 is considered a safe drug with few side effects. Studies in humans have not revealed an increase in blood clot risk [35,77,78].

## 5. Calcitonin

### 5.1. Mechanism of Action

Calcitonin is a peptide hormone secreted by the parafollicular cells of the thyroid gland. In clinical trials, salmon or eel calcitonin are widely used because of their enhanced potent activity compared to human calcitonin. Calcitonin binds to receptors mainly located on the membranes of osteoclasts, thus reducing their motility and bone resorption ability. It also prevents osteoclast precursors from maturing [79,80,81] (Figure 1). In addition, calcitonin possesses a pain relief effect. Ito et al. [82] demonstrated that chronic calcitonin treatment induced changes in serotonergic systems, which relieved hyperalgesia in ovariectomized rats. Normalization of the sodium channel and alleviation peripheral circulatory disturbance may also contribute to the analgesic effect [83,84].

### 5.2. Clinical Trials for the Treatment of Osteoporosis

The bone mass-increasing effect of eel calcitonin has been demonstrated in several RCTs [39,40]. A phase III clinical trial including 565 women with postmenopausal osteoporosis showed increases in lumbar, trochanteric, and femur BMD from baseline (1.5%, 0.6% and 0.3%, respectively), and decreases in bone resorption marker levels [41]. Combination of calcitonin with vitamin D_3_ or estrogen seems to further enhance the bone mass-increasing effect (3.44% at 12 months in the combination group vs 1.40, 0.92, and −0.70% in the calcitonin alone, 1α-hydroxycholecalciferol alone, and control groups, respectively) [40]. Only one RCT showed that the use of eel calcitonin would decrease the incidence of vertebral fractures by as high as 59% (RR, 0.41; 95% CI, 0.26–1.12) [42]. Two RCTs demonstrated that salmon calcitonin reduces the vertebral fracture rate in patients with postmenopausal osteoporosis [43,44].

Use of calcitonin significantly reduced osteoporotic bone pain compared to placebo at 6 months in a meta-analysis (standardized mean differences, −0.49; 95% CI, −0.85 to −0.13) [45,46]. Additionally, quality of life reportedly improved at least in part owing to the analgesic effect of calcitonin [47,48]. In a RCT in osteoporotic patients who underwent total hip replacement after fracture of the proximal femur, Peichl et al. [49] showed significant improvements in pain and activity of daily living in the group treated with salmon calcitonin, calcium, and vitamin D compared to the control group treated with calcium and vitamin D alone. Taken together, calcitonin may be a preferred treatment for acute osteoporotic fractures.

### 5.3. Adverse Events

The most common side effects involve nausea, reduced appetite, diarrhea, abdominal pain, and discomfort. Calcitonin may also contribute to hypocalcemia through its mechanism of action. Careful monitoring of the serum calcium level is required, and vitamin D and calcium supplementation may be beneficial for preventing hypocalcemia.

A systematic review of 21 RCTs showed an increased incidence of malignancy in patients treated with calcitonin (4.1%) when compared to patients given placebo (2.9%) [85]. On the other hand, Sondergaard et al. [86] reported that no increased risk of cancer in transgenic mice overexpressing salmon calcitonin. A recent review suggested that there is no reasonable connection between an oncogenic effect and the current understanding of the mechanism of action of calcitonin [87]. In a clinician’s guide, a weak association between malignant tumor, especially, basal cell carcinoma, and chronic use of intranasal calcitonin has been indicated [88]. Even in the case of a tiny possibility of oncogenic effect, one should be clinically prudent when prescribing this drug, and such possibility will likely influence physicians’ decisions.

## 6. Estrogen and Selective Estrogen Receptor Modulators

### 6.1. Mechanism of Action

Estrogen deficiency in postmenopausal women has been recognized as a potential risk of the impairment of bone formation since postmenopausal osteoporosis was reported in 1940 [6]. It has been suggested that bone remodeling is accelerated at menopause, based on increases in both markers of bone resorption and formation [89]. Therefore, contrary to the original hypothesis, the main contribution to bone loss in the setting of estrogen deficiency seems to be an increase in bone resorption, not impaired bone formation [90]. The recent development of mouse models with deletion of the estrogen receptor in specific cell types using the Cre/LoxP system has led to the identification of cellular targets of sex steroid action in vivo [91].

Selective ablation of estrogen receptor (ER) α in osteoclasts of females, but not males, led to an osteoporotic bone-like phenotype in postmenopausal women [92]. It was revealed that estrogen directly regulates the life span of mature osteoclasts via induction of the Fas/FasL system [92]. Deletion of ERα in osteoblasts had differential effects at different stages of differentiation of the osteoblast lineage. Almeida et al. [93] reported that deletion of ERα in mesenchymal progenitors or from osteoblast progenitors (using Prx1- or Osx1-Cre) caused a decrease in periosteal bone apposition and cortical bone mass. These effects resulted from the potentiation of Wnt/β-catenin signaling, which increases the proliferation and differentiation of periosteal osteoblast progenitor cells. On the other hand, deletion of ERα in osteoblasts and osteocytes expressing Col1a1- or Dmp1-Cre had no impact on bone mass and architecture [93,94]. Although the impact of estrogen on osteoblast lineage has not been fully elucidated, one of its most important downstream mediators is the OPG/RANKL system [95]. In general, estrogen seems to positively control bone volume.

### 6.2. Clinical Trials for the Treatment of Osteoporosis

Estrogen treatment is effective in the prevention and treatment of postmenopausal osteoporosis; however, serious side effects, such as cardiovascular events and breast cancer risk, were reported in the Women’s Health Initiative study, which recruited 16,609 postmenopausal women (HR for cardiovascular events, 1.29 (95% CI, 1.02–1.63); HR for breast cancer, 1.26 (1.00–1.59)) [96]. Due to these complications, estrogen is used for the prevention of postmenopausal symptoms only in the short term. SERMs activate distinct tissue receptors for estrogen. Raloxifene is a US Food and Drug Administration (FDA)-approved drug for the treatment of osteoporosis, and Ettinger et al. [50] demonstrated that raloxifene increased BMD slightly and decreased the risk of vertebral fractures (RR, 0.7; 95% CI, 0.5–0.8), but had no effect on non-vertebral fractures (RR, 0.9; 95% CI, 0.8–1.1) in the Multiple Outcomes of Raloxifene Evaluation trial. The same study revealed that raloxifene increased BMD in the femoral neck by 2.1% and in the spine by 2.6% compared to placebo. Recently, the combination therapy of raloxifene and bazedoxifene, another SERM, was approved by the FDA only to prevent, not to treat osteoporosis [97].

### 6.3. Adverse Events

A systematic review reported that hot flashes (OR, 1.58; 95% CI, 1.35–1.84), thromboembolic events (OR, 1.63; 95% CI, 1.36–1.98), pulmonary embolism (OR, 1.82; 95% CI, 1.16–2.92), and fatal strokes (OR, 1.56; 95% CI, 1.04–2.39) have been associated with raloxifene use compared with placebo [98]. Unlike hormone replacement therapy, long term use of raloxifene decreases breast cancer risk but increases the risk of thromboembolic events [99].

## 7. Bisphosphonates

### 7.1. Mechanism of Action

Etidronate was the first BP to be clinically used, in a child with fibrodysplasia ossificans progressiva [100]. BPs have been used to suppress bone resorption only since the 1990s [101], but they are currently the most common medications for osteoporosis [98]. BPs are chemically stable analogs of inorganic pyrophosphate and have a core structure made up of P-C-P bonds, which are responsible for the strong binding affinity toward hydroxyapatite [102]. This adsorption to bone surface enables BPs to make close contact with osteoclasts and inhibit bone resorption (Figure 1). BPs are classified into two groups; non-nitrogen-containing and nitrogen-containing. The former group comprises etidronate, clodronate, and tiludronate, with structures close to that of pyrophosphate. Non-nitrogen-containing BPs, which is first generation BPs, inhibit mitochondrial adenosine triphosphate translocases and induce apoptosis in osteoclasts [103]. In contrast, nitrogen-containing BPs, which have more potent anti-resorption activity than non-nitrogen-containing BPs, comprise pamidronate, alendronate, risedronate, ibandronate, and zoledronate [104]. Their higher anti-resorption potency results from their affinity for bone mineral and their ability to inhibit osteoclast function. Nitrogen-containing BPs target osteoclast farnesyl pyrophosphate synthase, which is the key enzyme of the mevalonate pathway, and blocks protein prenylation, thereby inhibiting bone resorption [105].

### 7.2. Clinical Trials for the Treatment of Osteoporosis

Etidronate was the first BP to enter clinical trials [101], followed by alendronate [106]. Three more BPs have been introduced since then, i.e., risedronate, ibandronate, and zoledronate [97]. Minodronate is developed and used in Japan [102]. Alendronate and risedronate are the most commonly used for the treatment of osteoporosis. A recent review reported that the fracture-protective effect of alendronate started at 12, 18, or 24 months after treatment for vertebral bone, hip, and non-vertebral bone, respectively [107].

A systematic review with network meta-analysis compared the efficacy of different BPs, including alendronate, clodronate, ibandronate, minodronate, pamidronate, risedronate, zoledronic acid, etidronate, and tiludronate, in short-term (2–5 years) fracture prevention for primary osteoporosis. Alendronate or zoledronate seemed to be the most effective in preventing hip fracture with network analysis in comparison to placebo (OR in alendronate, 0.60 (95% CI, 0.39–0.94); OR in zoledronate, 0.61 (95% CI, 0.48–0.79)). Additionally, zoledronate seemed the most effective in preventing vertebral fracture, non-vertebral fracture, and wrist fracture (OR for vertebral fracture, 0.34 (95% CI, 0.26–0.44); OR for non-vertebral fracture, 0.69 (95% CI, 0.61–0.79); OR for wrist fracture, 0.61 (95% CI, 0.48-0.79)) [51]. Zhou et al. [52] reported systematic review about the differential effect of BPs on prevention of fracture in men with osteoporosis. They found no significant difference between any pairs of alendronate, ibandronate, risedronate, and zoledronate for both vertebral and non-vertebral fractures (OR in alendronate, 0.22 (95% CI, 0.03–1.55) for vertebral fractures and 0.78 (95% CI, 0.13–4.65) for non-vertebral fractures; in ibandronate, 0.26 (95% CI, 0.02–4.25) and NA; in risedronate, 2.47 (95% CI, 0.09–69.00) and 0.53 (95% CI, 0.10–2.99); in zoledronate, 0.23 (95% CI, 0.05–1.06) and 0.63 (95% CI, 0.11–3.37)), though zoledronate ranked as the most effective for preventing vertebral fractures and risedronate ranked best for preventing non-vertebral fractures.

### 7.3. Adverse Events

The most commonly reported side effects for oral BPs are gastrointestinal disturbances because of their irritability to the gastrointestinal tract [107]. Etidronate, which is the only BP to be used as an inhibitor of ectopic calcification and ossification, decreases not only bone resorption but also calcification, and thus its long-term use is a potential risk for osteomalacia [108]. Other adverse events, such as BP-related osteonecrosis of the jaw, atypical subtrochanteric femoral fractures, arterial fibrillation, and acute renal failure, are extremely rare [109,110]. The pathology of BP-related osteonecrosis of the jaw is still unclear. However, Santini et al. [111] reported that BPs could lead to osteonecrosis through their effects on blood vessels in the bone by inhibiting vascular endothelial growth factor.

## 8. Anti-Receptor Activator of Nuclear Factor κB Ligand Antibody

### 8.1. Mechanism of Action

RANKL is an essential mediator in the bone resorption process. RANKL, which is expressed on osteoblasts, activates osteoclastogenesis by binding to RANK on the precursors of osteoclasts. Moreover, by binding to RANK of mature osteoclasts, RANKL stimulates bone resorption and inhibits osteoclast apoptosis [112,113] (Figure 1).

Denosumab is a fully human monoclonal antibody that binds to RANKL, thereby inhibiting the osteoclast-mediated bone resorption. Usually, denosumab is subcutaneously administered at a dosage of 60 mg every 6 months in the treatment of osteoporosis in postmenopausal women [114]. According to an off-treatment study, the effects of denosumab, such as increased BMD and decreased bone turnover markers, are reversible after the discontinuation of denosumab, which indicates that the biological activity of denosumab is sustained by its continuous administration [115].

### 8.2. Clinical Trials for the Treatment of Osteoporosis

In a meta-analysis, denosumab induced greater increases than did alendronate in BMD at the lumbar spine (weighted mean differences (WMD), 0.66; 95% CI, 0.56–0.75), total hip (WMD, 0.63, 95% CI, 0.46–0.79), femoral neck (WMD, 0.63; 95% CI, 0.57–0.69), and distal radius (WMD, 0.65; 95% CI, 0.17–1.13) [53]. Similarly, in a randomized, double-blinded, placebo-controlled trial (DIRECT), lumbar spine, total hip, femoral neck, and distal one-third radius BMD increased more in the denosumab group (9.1%, 4.6%, 4.0%, and 0.5%, respectively) than in the placebo group (0.1%, −1.1%, −1.1%, and −1.8%, respectively) after 24 months of treatment [54].

There are also many studies about the risk of fracture. A meta-analysis identified that denosumab administration resulted in a 42% fracture risk reduction as compared with placebo (RR, 0.58; 95% CI, 0.52–0.66) [55]. However, in another meta-analysis, no significant difference was noted in the reduction of fracture risk between denosumab and alendronate administrations (OR, 1.42; 95% CI, 0.84–2.40) [53]. In the Fracture Reduction Evaluation of Denosumab in Osteoporosis Every 6 Months (FREEDOM) study, denosumab significantly reduced the risk of vertebral, nonvertebral, and hip fractures (68%, 20%, and 40%, respectively) as compared to placebo at 36 months [56]. Similarly, in the DIRECT study, denosumab induced a 65.7% reduction in new or worsening vertebral fracture risk in comparison with placebo at 24 months of treatment [54].

### 8.3. Adverse Events

Several significant and serious adverse events, including hypocalcemia, osteonecrosis of the jaw, and atypical femoral fracture have been reported in the literature, and these incidences do not differ between denosumab and BPs [53,54,116,117,118]. The FREEDOM extension study evaluated the effect of 3 or 6 years of denosumab treatment; 6 out of 4550 participants experienced events of osteonecrosis of the jaw, whereas one participant developed an atypical femoral fracture. However, these incidences were low, and 6-year treatment with denosumab was well-tolerated in women with osteoporosis [119].

## 9. Parathyroid Hormone

### 9.1. Mechanism of Action

PTH is an 84-amino-acid polypeptide that is secreted by the parathyroid gland mainly in response to a low blood calcium level. The hormone adjusts serum calcium and phosphate. Binding of PTH to osteoblasts increases their RANKL expression. As a result, PTH indirectly increases osteoclast differentiation and function. Consequently, PTH enhances calcium release by bone resorption. Thus, PTH is important for bone remodeling.

Teriparatide (PTH1-34) consists of the first 34 N-terminal amino acids of PTH and is the biologically active region on the skeleton. It is well known that continuous PTH1-34 dosing results in a catabolic effect and, conversely, intermittent administration promoted an anabolic effect on bone [120]. The reason for these differential effects remains unclear. However, intermittent administration reportedly leads to expression of interleukin-11, which in turn suppresses DKK-1 and, eventually, activates Wnt signaling [121] (Figure 1).

Abaloparatide (PTHrP1-34) is an analog of PTH-related protein, which has been identified to cause humoral hypercalcemia of malignancy and shares homology with PTH1-34. PTHrP1-34 acts through not only PTH-receptor1 but also PTHrP-specific receptor [122]. Additionally, there are two high-affinity PTH-receptor1 conformations, R0 and RG, and PTHrP1-34 selectively binds to RG, which results in a more transient response than that induced via R0. Therefore, PTHrP1-34 is expected to induce a stronger anabolic effect than PTH1-34 [123].

### 9.2. Clinical Trials for the Treatment of Osteoporosis

Clinical trials of teriparatide have indicated positive results in osteoporosis. In the Fracture Prevention Trial, iliac crest bone biopsy specimens of postmenopausal women with osteoporosis (placebo (*n* = 19), 20 µg of teriparatide (*n* = 18), 40 µg of teriparatide (*n* = 14) per day, mean treatment time of 19 months) were analyzed. Teriparatide significantly increased cancellous bone volume (teriparatide, 14%; placebo, –24%) and reduced bone marrow star volume (teriparatide, –16%; placebo, 112%) as indicated by 2D histomorphometric analysis [57]. In a 21-month trial including 1637 postmenopausal women with a history of vertebral fractures, 20 µg of daily teriparatide increased BMD by 9 more percentage points in the lumbar spine and reduced the risk of new vertebral fractures by 65% compared with placebo [58].

Several clinical trials of abaloparatide have been recently reported. In a multicenter, multinational, double-blind placebo trial including 222 postmenopausal women with osteoporosis, increases in lumbar spine BMD were similar between 80 µg of daily abaloparatide and 20 µg of daily teriparatide (6.7% and 5.5%, respectively) 24 weeks after initial administration. Femoral neck BMD was increased by 3.1% by abaloparatide and by 1.1% by teriparatide, respectively. Moreover, total hip BMD increased more strongly after abaloparatide than after teriparatide treatment (2.6% vs. 0.5%) [59]. In the Abaloparatide Comparator Trial In Vertebral Endpoints (ACTIVE), which was a phase 3 trial that included 2463 postmenopausal women with low BMD, the effects of blinded daily subcutaneous injections of placebo (*n* = 821); abaloparatide 80 µg (*n* = 824); or open-label teriparatide 20 µg (*n* = 818) for 18 months were analyzed. The number of patients with new fractures (both vertebral and non-vertebral) were similar between abaloparatide and teriparatide (vertebral (abaloparatide, 0.6%; teriparatide, 0.8%; placebo, 4.2%), non-vertebral (abaloparatide, 2.7%; teriparatide, 4.7%; placebo, 3.3%)) [60]. Eighteen months of abaloparatide treatment followed by 6 months of alendronate treatment was reported as an extension of ACTIVE, which compared abaloparatide with placebo both followed by alendronate. In this study, use of abaloparatide followed by alendronate improved BMD and reduced fracture risk [61].

### 9.3. Adverse Events

Teriparatide and abaloparatide have similar adverse events. High doses of teriparatide induced an increased risk of osteosarcoma in growing rodents [124]. Likewise, abaloparatide dose- and time-dependently induced osteosarcoma formation in rats [125]. Therefore, the FDA has limited the treatment duration with teriparatide and abaloparatide to 24 months. However, a US postmarketing surveillance study did not detect an increase in osteosarcoma risk compared to that expected in the general population [126]. Other common adverse effects include muscle cramps, headache, and nausea. With regard to hypercalcemia, which is one of the adverse effects of PTH, the incidence was lower with abaloparatide (3.4%) than with teriparatide (6.4%) (risk difference, −2.96; 95% CI, −5.12 to −0.87; *p* = 0.006) [60].

## 10. Sclerostin Inhibitors

### 10.1. Mechanism of Action

The Wnt signaling pathway plays important roles in osteoblast differentiation by regulating the transcription of Runx2 [8,121,127]. After binding of Wnt to the low-density LRP-5/6 receptor on the osteoblast cell membrane [128,129,130], Wnt signaling stimulates osteoblast differentiation by activating of Runx2 through either β-catenin stabilization or protein kinase Cδ. Therefore, Wnt-signaling inhibitors, such as sclerostin and DKK-1, are promising for osteoporosis treatment. Sclerostin, which is the product of the *SOST* gene, is considered to be mainly produced by osteocytes, though sclerostin mRNA has been also detected in chondrocytes and several viscera [131,132,133,134,135,136]. The main function of sclerostin in bone metabolism lies in that it inhibits the Wnt/β-catenin pathway in osteoblasts via competitive binding of LBP-5/6, and thus, inhibits osteoblast differentiation [128,129,130]. Thus, sclerostin inhibition can induce osteoblast activation and promote bone formation (Figure 1).

### 10.2. Clinical Trials for the Treatment of Osteoporosis

Several anti-sclerostin antibodies have been developed for the treatment of osteoporosis in humans [62,137,138]. Romosozumab (AMG785: a humanized monoclonal anti-sclerostin antibody) has been clinically evaluated and was recently approved for clinical treatment of osteoporosis. In a phase 2 RCT in postmenopausal osteoporotic women, participants who received 210 mg romosozumab monthly over a 12-month period exhibited the largest increase of 11.3% in BMD from baseline at the lumbar spine, 4.1% at the total hip, and 3.7% at the femoral neck, among five dosing regimens of subcutaneous romosozumab [63]. These increases were significantly larger than those observed in alendronate and teriparatide treatment groups. Increase in serum procollagen type I N-terminal propeptide was transitory and continued up to one month after the initial administration, followed by a return to baseline at two months after the initial administration. In contrast, the serum level of β-isomer of the C-terminal telopeptide of type I collagen decreased initially after administration and remained below the baseline for 12 months [63]. A phase 3 clinical trial (STRUCTURE) in postmenopausal women with osteoporosis transitioning from oral BP to either romosozumab (210 mg/month) or teriparatide (20 µg/day) also showed stronger increases in BMD at all of the lumbar spine, total hip, and femoral neck at 6 months and 12 months after romosozumab administration than after teriparatide administration [64]. A phase 3 clinical trial (FRAME) in postmenopausal women with osteoporosis revealed that 12-month romosozumab administration (210 mg/month) reduced the risk of new vertebral fractures by 73% (16 out of 3321 patients (0.5%)) as compared with placebo (59 out of 3322 (1.8%)) [62]. However, romosozumab did not reduce the risk of non-vertebral fractures as compared with placebo. Another phase 3 clinical trial (ARCH) in postmenopausal women with osteoporosis also revealed that romosozumab treatment for 12 months followed by alendronate for 12 months resulted in a 48% reduction of new vertebral fracture risk and a 19% reduction of new non-vertebral fracture risk as compared with alendronate alone [65].

DKK-1 is another Wnt antagonist that blocks binding of Wnt proteins to LRP-5/6. The human anti-DKK-1 antibody BHQ880 is expected to be a promising target for skeletal-related events in multiple myeloma because DKK-1 is expressed by multiple myeloma cells and plays an important role in osteolytic bone disease [139,140]. In a recent study, sclerostin inhibition induced an increase in DKK-1 as negative feedback, which reduced the anabolic effect of sclerostin blockage in an animal model [141]. Therefore, the use of bispecific antibodies against both sclerostin and DKK-1 is expected to promote new bone formation more than monotherapy with an anti-sclerostin antibody, even in patients with osteoporosis. However, there have been no clinical trials of DKK-1 antagonist and the bispecific antibodies for osteoporosis treatment.

### 10.3. Adverse Events

In the largest clinical trial enrolling 7180 postmenopausal women (FRAME study), injection site reactions were observed in 5.2% of patients treated with romosozumab and in 2.9% of patients who received a placebo [62]. The frequencies of mortality and serious adverse events were balanced between the romosozumab and placebo groups [62]. However, the incidence of cardiac ischemic events was greater in the romosozumab group than in the alendronate group (OR, 2.65; 95% CI, 1.03–6.77) in the ARCH trial. The reason for this discrepancy and the mechanism underlying cardiac ischemic events are still unclear [65]. The incidences of osteonecrosis of the jaw and atypical femoral fracture were not significantly increased in the romosozumab group as compared with the placebo or alendronate groups in the FRAME and ARCH studies, though these clinical trials were based on only two-year observational periods [62,65].

## 11. Combination Therapy

### 11.1. Mechanism of Action

Combination therapy with different therapeutic agents is potentially more efficacious than treatment with either agent alone. An ideal anti-osteoporosis agent increases new bone formation and simultaneously inhibits bone resorption [142]; however, there are currently no anti-resorptive agents, such as BPs, denosumab, and SERMs, or no anabolic agents, such as PTH and its analogs, which meet both therapeutic goals. Recently, combinations of anabolic agents and anti-resorptive agents are attracting interest because they are expected to be an ideal anti-osteoporosis option. In a recent systematic review and meta-analysis of RCTs evaluating the efficacies of combination therapies with PTH analogs and anti-resorptive agents including BPs, denosumab, and estrogen-like drugs, combination therapy exhibited superior efficacy over monotherapy with an additional 36% reduction in fracture risk [143].

### 11.2. Clinical Trials for the Treatment of Osteoporosis

Numerous clinical trials have evaluated the efficacy of combinations of anabolic and anti-resorptive agents (Table 2).

#### 11.2.1. Combination of BPs and Anabolic Agents

The synergistic efficacy of BPs and anabolic agents is controversial. Early clinical trials reported that there was no evidence of synergy between BPs and PTH analogs in postmenopausal and male osteoporosis [144,145,146]. Other studies suggested that BPs can attenuate PTH-induced modeling-based bone formation [144,145]. Cosman et al. [147] reported that a combination of teriparatide and intravenous zoledronic acid increased BMD more rapidly than either agent alone (13 and 26 weeks, respectively; *p* < 0.001). In a randomized double-blinded study of male osteoporosis, combination therapy with teriparatide and risedronate provided a greater increase in total hip BMD than either monotherapy (3.86%, 0.29% and 0.82%, respectively; *p* < 0.05 for both) [148].

#### 11.2.2. Combination of Denosumab and Anabolic Agents

The efficacy of combination therapy with denosumab and anabolic agents has been evaluated in both animals and humans [149,150,151,152,153]. In a RCT of postmenopausal osteoporosis with 24 months of treatment, lumbar spine, femoral neck, and total hip BMD increased more with a combination of denosumab and teriparatide (12.9%, 6.8%, and 6.3%, respectively) than with denosumab (8.3%, 4.1%, and 3.2%, respectively) or teriparatide (9.5%, 2.8%, and 2.0%, respectively) monotherapy [152]. Although these studies did not assess fracture risk reduction, the combination of teriparatide and denosumab may be useful to treat patients at very high risk of fragility fracture [151,152,153].

#### 11.2.3. Combination of SERMs and Anabolic Agents

A few studies evaluated a combination of a SERM and an anabolic agent [154,155]. In patients who had received prior treatment of teriparatide for 9 months, switching to combination therapy with raloxifene plus teriparatide had superior efficacy compared to the continuation of teriparatide monotherapy for increasing lumbar spine BMD for another 9 months (6.0% and 2.8% respectively, *p* = 0.032) [155]. Conversely, in patients on prior raloxifene treatment for at least 18 months, there was no significant difference in increase in lumbar spine BMD between patients switched to combination therapy with raloxifene plus teriparatide and teriparatide monotherapy (9.2% and 8.1%) [154].

### 11.3. Adverse Events

There is concern that combination therapies may increase the risk of serious adverse events when compared to monotherapies. When combining anabolic agents with BPs [144,145,146,147,148], denosumab [151,152], or SERMs [154,155], most studies reported no significant increase in the frequency of serious adverse events when compared to treatment with each agent alone.

## 12. Stem Cells

### 12.1. Mechanism of Action

Stem cell therapy has received increasing interest for the treatment of various diseases. Stem cells possess self-renewal and plasticity, and therefore, can repair and renew damaged tissue. Thus, stem cells are believed to be the best source for cell replacement therapy of bone disease. There are various types of stem cells, including embryonic stem (ES) cells, induced pluripotent stem (iPS) cells, and somatic stem cells such as MSCs. As MSCs are easy to isolate and the use of ES cells and iPS cells are laden with safety and ethical issues [156], MSCs have received the most attention for the treatment of osteoporosis. Osteoporosis is characterized by bone mass reduction related to the poor supply of bone marrow stromal cells, which are a heterogeneous population of cells. The mechanism of bone formation via MSC transplantation is considered to involve (1) MSCs directly covering the pathologic area and differentiating into osteoblasts and (2) MSCs secreting various growth factors that stimulate angiogenesis and decrease osteoclastic differentiation, thus indirectly contributing to repair of the damaged site [157,158,159].

### 12.2. Clinical Trials for the Treatment of Osteoporosis

MSCs, including bone marrow-derived (BM-)MSCs, adipose tissue-derived (AD-)MSCs, and umbilical cord-derived MSCs, have positive effects on osteoporosis in animal models [157,160,161]. Recently, the use of autologous BM-MSCs for the treatment of osteoporosis has been tested in a phase I clinical trial. Cells were collected 30 days before infusion and were cultured in good manufacturing practice conditions to establish a dose range. BM-MSCs were fucosylated and were intravenously infused into patients with osteoporosis. This clinical study is still in the process of recruiting (ClinicalTrials.gov identifier: NCT02566655).

One phase II clinical trial on the use of AD-MSCs for the treatment of proximal humeral fracture by low-energy trauma in patients over 60 years of age has been conducted. AD-MSCs were procured from the patients and were embedded in fibrin gel and then wrapped around hydroxyapatite granules. Radiological follow-up and functional assessment were performed. Unfortunately, the study was ended due to slow recruitment and no results were reported (ClinicalTrials.gov identifier: NCT01532076).

Derivation of MSCs from iPS cells (iPSC-MSCs) has been studied [162]. Qi et al. [163] reported that exosomes secreted by human iPS-MSCs repaired bone defect in a rat model of osteoporosis. In their study, human iPSC-MSCs exosomes were dropped onto β-tricalcium phosphate scaffold and implanted into the calvarial defects in ovariectomized rats. After 8 weeks, exosomes significantly stimulated bone regeneration and angiogenesis.

### 12.3. Adverse Events

As only a few clinical studies have been conducted, adverse events of stem cells for the treatment of osteoporosis are still unknown. However, a meta-analysis of 36 clinical studies including 1012 participants treated with mesenchymal stromal cell therapies for other diseases showed the safety of the stem cell therapies. There were no significant differences in the occurrence of acute infusional toxicity (OR, 2.12; 95% CI, 0.55–8.77) and long term adverse events (OR, 0.60; 95% CI, 0.28–1.25) between the mesenchymal stromal cell and control groups [164]. In addition, MSCs have an immunosuppressive property and are considered to be safe for clinical use by the FDA [165]. In contrast, iPS cells are associated with a risk of tumorigenesis [166]. Thus, for the use of iPS cells to treat osteoporosis, further studies are needed.

## 13. Conclusions and Future Directions

This review summarized the mechanisms of action, major clinical trials, and side effects for current therapeutics for osteoporosis. Anti-bone-resorptive agents have been widely used to date. In addition, bone anabolic agents have recently become available based on advanced clinical trials. Clinical use of these drugs has shed light on the mechanism of osteoporosis. To establish an ideal anti-osteoporosis therapy that increases new bone formation and simultaneously inhibits bone resorption, anabolic agents combined with anti-resorptive agents have been tested in several recent clinical trials.

The pathophysiology of osteoporosis varies among individuals; thus, treatment should be personalized. Identification of the individual pathology of bone metabolism will allow more personalized treatment with combination therapies. Moreover, the recent progress in bone pharmacophysiology will provide novel agents for the treatment of osteoporosis. Especially, cell and gene engineering techniques have tremendous potential to bolster innovation in drug delivery systems and the microenvironment for bone remodeling. In this view, MSCs have been widely tested for therapy [165,167]. iPS cells have been attracting great attention for osteoporosis treatment in recent years owing to their high pluripotency. However, iPS cells have tumorigenicity, and the safety of the use of iPS cells will probably remain a future study subject. Small interfering RNA (siRNA) therapies are one of the therapies based on gene engineering. SiRNAs can specifically knockdown genes related to bone metabolism, such as *RANKL/RANK*, *DKK-1*, and *SOST*, and thus, their bioactivity can be reversed over time. However, targeted systemic delivery of therapeutic amounts still proves difficult and siRNA off-target effects can be worrisome [168]. Further studies are needed for the clinical application of such newly developed agents.

There are many treatments for osteoporosis. Each treatment has benefits but also risks of side effects. It is important that clinicians have the right information about the therapies and select the best treatment for individual patients. Clinical studies particularly for the newly developed therapies are still not so much and provide little evidence due to the small sample size and short term follow-up. However, understanding of the mechanism of drug action partially helps us to make the right treatment strategy. To make a better treatment choice, further studies that reveal the clinical outcomes supporting the molecular mechanism of the drugs are needed.

## Figures and Tables

**Figure 1 ijms-20-02557-f001:**
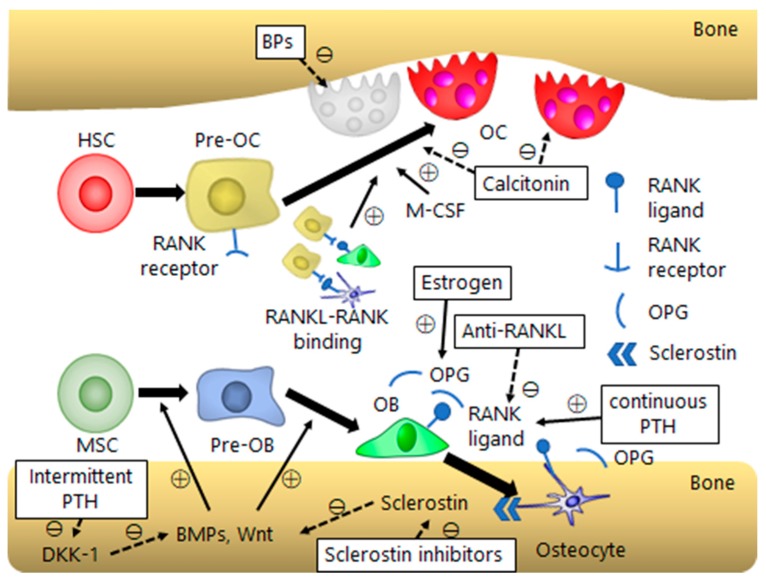
Schematic diagram of bone homeostasis and a summary of the action mechanism of the agents for osteoporosis. BP, bisphosphonate; DKK, dickkopf; M-CSF, monocyte/macrophage colony-stimulating factor; PTH, parathyroid hormone; normal arrows with “+” mean positive effect; dotted arrows with “−” mean negative effect. This figure is copyright free.

**Table 1 ijms-20-02557-t001:** Summary of the treatment osteoporosis identified in this review.

Agents	Mechanism of Action	Effect on Bone Metabolism	Side Effects	Clinical Trials and Meta-Analysis
Calcium	Reduction of PTH release	Inhibition of bone resorption	Gastrointestinal disorders Hypercalcemia	[14,15,16,17,18,19,20,21,22,23,24,25,26]
Vitamin D	Modulation of the calcium metabolism	Inhibition of bone resorption	Appears to be safe	[27,28,29,30,31,32]
Vitamin K2	Help of the osteocalcin γ-carboxylation	Activation of bone formation	Appears to be safe	[33,34,35,36,37,38]
Calcitonin	Regulation of osteoclast function Prevention of osteoclast precursors from maturing	Inhibition of bone resorption	Gastrointestinal disorders Hypocalcemia Weak association between malignant tumor	[39,40,41,42,43,44,45,46,47,48,49]
SERMs	Interaction with RANKL/RANK/OPG system	Inhibition of bone resorption	Thromboembolic events Pulmonary embolism Fatal strokes	[50]
Bisphosphonates	Induction of osteoclast apoptosis	Inhibition of bone resorption	Gastrointestinal disorders Osteonecrosis of the jaw Atypical femoral fractures Acute renal failure	[51,52]
Anti-RANKL antibody	Prevention of the RANKL/RANK system	Inhibition of bone resorption	Osteonecrosis of the jaw Atypical fracture Hypocalcemia	[53,54,55,56]
PTH	Stimulation of osteoblast differentiation	Activation of bone formation (intermittent PTH)	Hypercalcemia Increasing risk of osteosarcoma	[57,58,59,60,61]
Sclerostin inhibitors	Regulation of BMP and Wnt signaling	Activation of bone formation	Cardiac ischemic event	[62,63,64,65]
Stem cells	Differentiate into osteoblasts directly Secretion of various growth factors	Supplementation of cell source for osteoblasts	Appears to be safe	N/A

PTH, parathyroid hormone; SERMs, selective estrogen receptor modulators; RANKL, receptor activator of nuclear factor κB ligand; OPG, osteoprotegerin; N/A, not available; BMP, bone morphogenetic protein.

**Table 2 ijms-20-02557-t002:** Summary of combination therapy with anabolic and anti-resorptive agents.

Anabolic Agent	Anti-Resorptive Agent	Main Therapeutic Effect Compared to Effect of Either Therapy	References
**PTH (teriparatide)**	BP (alendronate, intravenous zoledronic acid)	More rapid BMD increase Greater increase in total hip BMD	[144,145,146,147,148]
	Anti-RANKL antibody (denosumab)	Greater increase in lumbar spine, femoral neck, total hip BMD	[151,152,153]
	SERM (raloxifene)	Greater increase in lumbar spine BMD	[154,155]

PTH, parathyroid hormone; BMD, bone mineral density; RANKL, receptor activator of nuclear factor κB ligand; SERM, selective estrogen receptor modulator.

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
