# Peer review of "Molecular-Based Treatment Strategies for Osteoporosis: A Literature Review"

_ijms, 2019, doi:10.3390/ijms20102557_

Round 1
Reviewer 1 Report
This is a beautifully written review about different treatment options for osteoporosis. The review covers era well the current treatment options and discusses potential future therapies.
Author Response
Point: This is a beautifully written review about different treatment options for osteoporosis. The review covers era well the current treatment options and discusses potential future therapies.
Response: Thank you very much for reviewing our manuscript. We hope that this manuscript aid clinicians to decide how to select the best treatment option. Thank you again.
Reviewer 2 Report
The introduction to this review clearly explain the physiopathology of osteoporosis and the different possibilities of pharmacological treatment according to the two theoretical disease mechanisms implicated, that is the lower bone formation or the increased bone resorption leading to ostroporosis disease.
The review is well structured, easy to read and the topics covered for each treatment (the mechanism of Action, the clinical evidences based in different clinical trials and meta-analyses and the adverse events) are ordered making easier the pros and cons of every treatment.
Bibliography used is up to date.
Author Response
Point:The introduction to this review clearly explain the physiopathology of osteoporosis and the different possibilities of pharmacological treatment according to the two theoretical disease mechanisms implicated, that is the lower bone formation or the increased bone resorption leading to ostroporosis disease. The review is well structured, easy to read and the topics covered for each treatment (the mechanism of Action, the clinical evidences based in different clinical trials and meta-analyses and the adverse events) are ordered making easier the pros and cons of every treatment. Bibliography used is up to date.
Response:Thank you very much for reviewing our manuscript. We hope that this manuscript aid clinicians to decide how to select the best treatment option. Thank you again.
Reviewer 3 Report
The topic of the review by Yiuchiro Ukon et al. is very interesting and the work seems well organized.
To give much more value to this extensive review, the Auhtors should specify their publication search strategies to select reported meta-analyses, clinical trials, reviews and others, adding also a flowchart showing the selection process. Otherwise it is not possible to check and evaluate the bias on the information reported in this review deriving from having considered only certain publications, with the possible consequent loss of value of the review itself.
In my opinion, each 'Clinical Trials for the Treatment of Osteoporosis' and 'Adverse Events' section should have the same structure, reporting firstly information derived by meta-analyses and then those by clinical trials, review and others. For meta-analyses, it is mandatory to present results by reporting also the numeric value of estimators, providing their scientific significance in the context of each meta-analyses. The results of clinical trials shoud be reported as the Authors did for 'Parathyroid Hormone' and 'Sclerostin inhibitors'. As well as, the search strategies to select ended and ongoing clinical studies on ClinicalTrials.gov should be reported.
Regarding Conclusions and Future Directions, the Authors should summarize findings and suggest a 'take-home' message. Some considerations should be done on the necessity of further clinical studies overcoming some limitations that the Authors of selected clinical trials had surely stated in their works.
I suggest to remove drugs and treaments from Figure 1, because they do not take part to bone homeostasis or to change le figure legend.
Author Response
Point 1: The topic of the review by Yiuchiro Ukon et al. is very interesting and the work seems well organized. To give much more value to this extensive review, the Authors should specify their publication search strategies to select reported meta-analyses, clinical trials, reviews and others, adding also a flowchart showing the selection process. Otherwise it is not possible to check and evaluate the bias on the information reported in this review deriving from having considered only certain publications, with the possible consequent loss of value of the review itself. As well as, the search strategies to select ended and ongoing clinical studies on ClinicalTrials.gov should be reported.
Response 1: Thank you for your kind suggestion. Our present review article is not “systematic or meta-analysis reviews” but a “narrative (literature)” review. We agree with your opinion if this article would be focused on the effectiveness of various drugs for osteoporosis, but we mainly focused on the molecular action mechanisms of various drugs in this article. Thus, we did not apply a systematic or meta-analysis review fashion. We would appreciate your understanding and we added the word as follows:
p3
In this literature review, we summarize the mechanisms of action of current and anticipated drugs in terms of basic bone biology.
Point 2: In my opinion, each 'Clinical Trials for the Treatment of Osteoporosis' and 'Adverse Events' section should have the same structure, reporting firstly information derived by meta-analyses and then those by clinical trials, review and others.
Response 2: Thank you for your kind suggestion. We correct the structure of each paragraph. The structure is firstly information derived by meta-analysis, then clinical trials, review and others.
Point 3: For meta-analyses, it is mandatory to present results by reporting also the numeric value of estimators, providing their scientific significance in the context of each meta-analyses. The results of clinical trials should be reported as the Authors did for 'Parathyroid Hormone' and 'Sclerostin inhibitors'.
Response 3: Thank you for your kind suggestion. We added the numeric values of estimators to the reference in the text especially for meta-analysis. And the results of clinical trials were also changed as we did for 'Parathyroid Hormone' and 'Sclerostin inhibitors'.
Point 4: Regarding Conclusions and Future Directions, the Authors should summarize findings and suggest a 'take-home' message. Some considerations should be done on the necessity of further clinical studies overcoming some limitations that the Authors of selected clinical trials had surely stated in their works.
Response 4:
Thank you for your kind suggestion. We added the word to conclusion and future direction as follows;
p17
There are many treatments for the osteoporosis. Each treatment has benefits but also risks of side effects. It is important that clinicians have the right information about the therapies and select the best treatment for the individual patients. Clinical studies particularly for the new developed therapies are still not so much and provide little evidence due to the small sample size and short term follow-up. However, understanding of the mechanism of drug action partially helps us to make the right treatment strategy. To make better treatment choice, further studies that reveal the clinical outcomes supporting the molecular mechanism of the drugs are needed.
Point 5: I suggest to remove drugs and treatments from Figure 1, because they do not take part to bone homeostasis or to change the figure legend.
Response 5:
Thank you for your kind suggestion. We changed the figure legend as follows;
p2
Figure 1. Schematic diagram of bone homeostasis and mechanism of action of the agents for osteoporosis.
Thank you very much for reviewing our manuscript.
We have earnestly appreciated your suggestions and hope that the revised manuscript meets your standards. Thank you again.
Round 2
Reviewer 3 Report
Point 1
I understand the aim of your literature review and I agree your added sentence. I suggest you to specific as well in the title this aspect like "Molecular-based Treatment Strategies for Osteoporosis: a literature review".
Author Response
Thank you very much for your valuable recommendation for the title of our manuscript. We agree with your suggestion and change the title as you recommended (Molecular-based Treatment Strategies for Osteoporosis: a literature review). Thank you.